# Generative Adversarial Training with Perturbed Token Detection for Model Robustness

**Jiahao Zhao, Wenji Mao**[*]

Institute of Automation, Chinese Academy of Sciences
School of Artificial Intelligence, University of Chinese Academy of Sciences
{zhaojiahao2019,wenji.mao}@ia.ac.cn

## Abstract

Adversarial training is the dominant strategy towards model robustness. Current adversarial training methods typically apply perturbations to embedding representations, whereas actual text-based attacks introduce perturbations as discrete tokens. Thus there exists a gap between the continuous embedding representations and discrete text tokens that hampers the effectiveness of adversarial training. Moreover, the continuous representations of perturbations cannot be further utilized, resulting in the suboptimal performance. To bridge this gap for adversarial robustness, in this paper, we devise a novel generative adversarial training framework that integrates gradient-based learning, adversarial example generation and perturbed token detection. Our proposed framework consists of generative adversarial attack and adversarial training process. Specifically, in generative adversarial attack, the embeddings are shared between the classifier and the generative model, which enables the generative model to leverage the gradients from the classifier for generating perturbed tokens. Then, adversarial training process combines adversarial regularization with perturbed token detection to provide token-level supervision and improve the efficiency of sample utilization. Extensive experiments on five datasets from the AdvGLUE benchmark demonstrate that our framework significantly enhances the model robustness, surpassing the state-of-the-art results of ChatGPT by 10% in average accuracy.

## 1 Introduction

Pre-trained language models (PLMs) and large language models (LLMs) (Ouyang et al., 2022) have made remarkable advancements in natural language processing (NLP). Given their increasing impact on science, society and individuals, adversarial robustness has become a crucial task for building trustworthy NLP systems (Kaur et al., 2022).

The adversarial vulnerability of deep learning models is a long-standing problem (Goodfellow et al., 2015). Various attack methods have demonstrated that even LLMs can be deceived with small, intentionally crafted perturbations (e.g., typos and synonym substitution) (Jin et al., 2020; Li et al., 2020; Chen et al., 2021; Wang et al., 2022; Liu et al., 2022a; Wang et al., 2023b). In response to adversarial attacks, many adversarial defense methods have been proposed to enhance model robustness. Among them, adversarial training (Miyato et al., 2019; Wang and Bansal, 2018; Zhu et al., 2020; Wang et al., 2021c; Ni et al., 2022; Xi et al., 2022) is widely recognized as the most effective approach, which involves continuous perturbations. In adversarial training, the gradient-based perturbations are introduced at the embedding layer while solving a min-max problem. Another form of adversarial training is adversarial augmentation (Jin et al., 2020; Li et al., 2020; Si et al., 2021; Ivgi and Berant, 2021; Maheshwary et al., 2021; Liu et al., 2022a), which utilizes discrete perturbations. In adversarial augmentation, the search for adversarial examples is treated as a combinatorial optimization problem and the model is trained using the augmented adversarial examples.

However, existing adversarial training methods have encountered non-trivial challenges in real-world applications. In adversarial training, the perturbations applied at the embedding layer create a gap between the continuous perturbations used during training and the discrete perturbations that real attack applies in the testing phase (Xu et al., 2020). On the other hand, adversarial augmentation demands high computational cost, typically with hundreds to thousands of queries per example (Maheshwary et al., 2021). Consequently, the continuous representations of perturbations cannot be further utilized in adversarial training, leading to the suboptimal performance.

In this work, we aim to address the afore-

---

[*]Corresponding author

mentioned issues and make a substantial step towards adversarial robustness. We propose a novel **Gener**ative **A**dversarial **T**raining (**GenerAT**) framework that consists of generative adversarial attack and adversarial training process. To bridge the gap between continuous perturbations and discrete text tokens, generative adversarial attack generates discrete perturbed tokens based on gradients. Specifically, our gradient-based attack calculates adversarial gradients with the classifier through forward and backward propagations, which is more efficient than search-based adversarial augmentation. The accumulated gradients are then shared at the embedding layer between the classifier and the generative model, so that they can be leveraged to guide the perturbed token generation. Besides, to get more robust representations, our generative adversarial attack is built upon the discriminative PLM (He et al., 2023), which is capable of distinguishing subtle semantic differences between similar words compared to other PLMs like BERT that most defense methods concentrate on.

Adversarial training process utilizes adversarial regularization to further improve robustness, by restricting the representations between the original and corresponding adversarial examples. As the generated perturbed tokens can provide fine-grained token-level supervision, adversarial regularization is integrated with perturbed token detection in the training process to improve sample usage efficiency. We conduct robustness experiments on the challenging AdvGLUE benchmark (Wang et al., 2021b), which applies multiple types of adversarial attacks on five datasets from GLUE (Wang et al., 2018). Extensive experiments show that our framework significantly improves adversarial robustness on all the datasets, surpassing ChatGPT (Wang et al., 2023b) by 10% in average accuracy and establishing new state-of-the-art results.

The main contributions are as follows:

- We propose the first generative adversarial training framework for adversarial robustness. Based on discriminative PLM, our framework provides a comprehensive means to integrate gradient-based learning, adversarial example generation and perturbed token detection.

- The generative adversarial attack in our framework exploits gradient propagation through sharing embeddings between the classifier and the generator, which enables the generator to effectively generate perturbed tokens.

- The adversarial training process in our framework further combines adversarial regularization with perturbed token detection, which improves the efficiency of sample utilization.

- Extensive experiments on five datasets demonstrate the effectiveness of our generative adversarial training framework, which consistently improves robustness by a large margin.

## 2 Related Works

### 2.1 Adversarial Defense and Detection

Adversarial training is acknowledged as the most effective defense method. In text domain, adversarial training (Sato et al., 2018; Zhu et al., 2020; Li and Qiu, 2021; Wang et al., 2021a; Li et al., 2021; Ni et al., 2021; Pan et al., 2022) solves the min-max optimization problem and injects continuous adversarial perturbations in the embedding layer, which leaves a gap between real discrete text perturbations. Adversarial training in continuous representation is an approximation of the real discrete adversarial perturbations in text. Previously developed adversarial training methods try to improve the approximation (Li et al., 2021; Pan et al., 2022). However, this is a hard problem and the quality of the approximation significantly affects model robustness. Our generative adversarial training takes the approach to directly bridge the gap between continuous embedding representations and discrete text tokens, and generates adversarial replaced tokens based on gradients.

Adversarial augmentation (Jin et al., 2020; Li et al., 2020; Si et al., 2021; Ivgi and Berant, 2021; Maheshwary et al., 2021; Liu et al., 2022a) searches the most effective replacement of tokens to generate adversarial examples by solving combinational problem and then using the augmented examples to retrain the model. However, these methods need over hundreds of iterations per example to find the corresponding replacement which brings high computational cost. To tackle these challenges, we propose a novel generative adversarial training framework that leverages gradients for the generation of discrete perturbed tokens.

Another line of work is adversarial detection (Zhou et al., 2019; Mozes et al., 2021; Nguyen-Son et al., 2022; Li et al., 2023) which focuses on detecting perturbed tokens. These methods typically detect replaced tokens and subsequently restore them to their original form, allowing the model

to make predictions on clean, restored data. In contrast to these methods, in our framework, we incorporate perturbed token detection to provide token-level supervision and enhance efficiency in the training process.

## 2.2 Pre-trained Language Models

BERT (Devlin et al., 2019) is an encoder-only language model that was trained with the masked language modeling (MLM) task. It represents a significant milestone in transformer-based PLMs and reveals the great potential of PLMs. After that, diverse forms of PLMs have emerged. One category is decoder-only PLMs, such as GPT-3 (Brown et al., 2020), OPT (Zhang et al., 2022) and BLOOM (Scao et al., 2022). These auto-aggressive language models are trained with causal language modeling (CLM). Another category is encoder-decoder PLMs like T5 (Raffel et al., 2020) and FLAN-T5 (Chung et al., 2022). These models convert a variety of text-based language problems into text-to-text format and train the model as a sequence-to-sequence generation problem.

In contrast, discriminative PLMs, such as ELEC-TRA (Clark et al., 2020) and DeBERTa-v3 (He et al., 2023) have received comparatively less attention. A discriminative PLM contains two transformer encoders. One is the generator trained with MLM, and the other is the discriminator trained with replaced token detection (RTD) task, that determines whether the tokens are replaced by the generator. We argue that discriminative language models learn the representations by distinguishing the subtle semantic difference between similar words, leading to more robust representations compared to BERT-style context-based cloze completion tasks. Therefore, our framework is built upon discriminative PLMs. A detailed comparison between these models is provided in Appendix A.

## 3 Proposed Method

Adversarial training typically applies perturbations to embedding representations, whereas real-world adversarial examples introduce perturbations at discrete text tokens. To bridge the gap between continuous embeddings and discrete tokens, we propose a novel generative adversarial training framework GenerAT that consists of generative adversarial attack and adversarial training process. We present the overall design of our framework as well as each component in this section.

### 3.1 Preliminaries

**Problem Formulation** We consider a general text classification problem with dataset $\mathcal{D} = \{\boldsymbol{x_i}, y_i\}_{i=1}^N$ consisting of inputs text $\boldsymbol{x_i} \in \boldsymbol{X}$ and labels $y_i \in \mathcal{Y}$. Adversarial attack aims to deceive the victim model by generating adversarial examples. Our goal is to train the model on $\mathcal{D}$ to maximize adversarial robustness, that is, to increase model accuracy on adversarial examples.

**Discriminative Pre-trained Models** Discriminative PLMs such as ELECTRA (Clark et al., 2020) and DeBERTaV3 (He et al., 2023) differ from BERT (Devlin et al., 2019) in their pre-training task. Instead of masked language modeling (MLM), discriminative PLMs employ a replaced token detection (RTD) approach. In this setup, a generator and a discriminator are jointly trained. The generator, which is a transformer encoder, is trained on MLM to generate ambiguous tokens that substitute the original tokens in the input sequence. On the other hand, the discriminator, a larger transformer encoder, is trained as a token-level binary classifier to determine whether the tokens have been replaced by the generator. The discriminator is then used for fine-tuning in downstream tasks.

### 3.2 Generative Adversarial Attack

Adversarial attacks deceive the victim model by applying small but intentionally worst-case perturbations to the original input. For example, in FGSM (Goodfellow et al., 2015) attack, the adversarial gradients are used as the perturbation:

$$\boldsymbol{\eta} = \epsilon \operatorname{sign}\left(\nabla_{\boldsymbol{x}} J(\boldsymbol{x}, y)\right) \tag{1}$$

where $\boldsymbol{\eta}$ is the perturbation, $\epsilon$ is the hyperparameter that controls the perturbation size, and $J$ denotes the loss function used for training the victim model.

While pixels in an image are continuous values that can be directly added with perturbations, text data are discrete tokens in nature. As a result, existing gradient-based methods (Sato et al., 2018; Zhu et al., 2020; Li and Qiu, 2021; Li et al., 2021; Pan et al., 2022) apply adversarial gradients to embedding representation of text:

$$\boldsymbol{x_{adv}} = E(\boldsymbol{x}) + \boldsymbol{\eta} \tag{2}$$

where $\boldsymbol{x_{adv}}$ is the virtual adversarial example of $\boldsymbol{x}$, $E(\boldsymbol{x})$ is the embedding of $\boldsymbol{x}$.

However, actual text perturbations introduced by adversarial attacks, such as synonym replace-

Step 1: Calculate Adversarial Gradients       Step 2: Generate Adversarial Examples

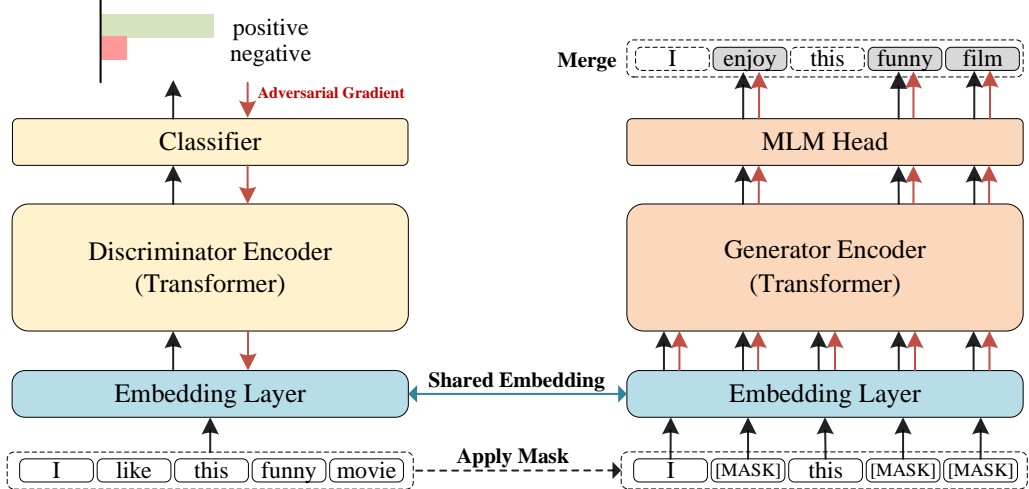

Figure 1: Overview of generative adversarial attack. The adversarial gradients are first calculated on the discriminator and accumulated to the embedding layer. Through the shared embedding with the generator, the generator leverages the gradients and generates perturbed tokens.

ment, are applied to discrete tokens. The gap between continuous perturbations and actual discrete tokens limits the effectiveness of adversarial training. Therefore, to bridge the gap, we introduce a novel generative adversarial attack.

As shown in Figure 1, we integrate one discriminator as the victim model and one generator to generate adversarial perturbed tokens. We first calculate the gradients on the victim model. The adversarial gradients are passed from the discriminator to the generator through the shared embedding layer. Then generator generates adversarial perturbed tokens with adversarial gradients.

**Calculate Adversarial Gradients** We calculate gradients on the discriminator. Input text $x$ is passed through the discriminator with a classifier header to calculate the loss. Using binary classification as an example, the loss function of the victim model is cross entropy:

$$J(\boldsymbol{x}, y) = -y \log \sigma(\boldsymbol{x}) - (1-y) \log(1 - \sigma(\boldsymbol{x}))$$
$$g = \nabla_{\boldsymbol{x}} J(\boldsymbol{x}, y)$$
$$(3)$$

where $\sigma(\cdot)$ denotes the sigmoid function, $g$ is the gradient respect to $(\boldsymbol{x}, y)$.

By integrating Eq.(3) with Eq.(1) and Eq.(2), the gradients are accumulated at the embedding layer. To enhance the training stability, we apply the perturbations to the normalized word embeddings.

**Generate Adversarial Examples** A generator is employed to generate perturbed tokens. The generator is a pre-trained language model with an MLM header that is able to generate semantically closed tokens of the masked tokens.

Similar to MLM, we first mask some percentage of the input tokens at random.

$$\boldsymbol{x}' = f_{\mathrm{mask}}(\boldsymbol{x}) \tag{4}$$

where $f_{\mathrm{mask}}(\cdot)$ is the mask strategy and $\boldsymbol{x}'$ is the masked sample.

Then the generator predicts the masked tokens. The probability for a particular masked token $x_t$ is:

$$p_{\theta_G}\left(x_t \mid \boldsymbol{x}' + \boldsymbol{\eta}\right) = \mathrm{softmax}\left(H_{\theta_G}(\boldsymbol{x}' + \boldsymbol{\eta})\right) \tag{5}$$

where $\boldsymbol{\eta}$ is the perturbation and $\theta_G$ denotes the parameter of the generator.

As mentioned before, we share the embedding layer between the discriminator and the generator. Consequently, gradients calculated on the discriminator are accumulated on the embedding layer and can propagate through the masked token prediction of the generator. In this way, we fill the gap between continuous gradients and discrete text tokens.

Lastly, we merge the generated token with the original tokens:

$$\hat{x}_i = \begin{cases} \hat{x}_i \sim p_{\theta_G}\left(x_i \mid \boldsymbol{x}' + \boldsymbol{\eta}\right), & i \in \mathcal{C} \\ x_i, & i \notin \mathcal{C} \end{cases} \tag{6}$$

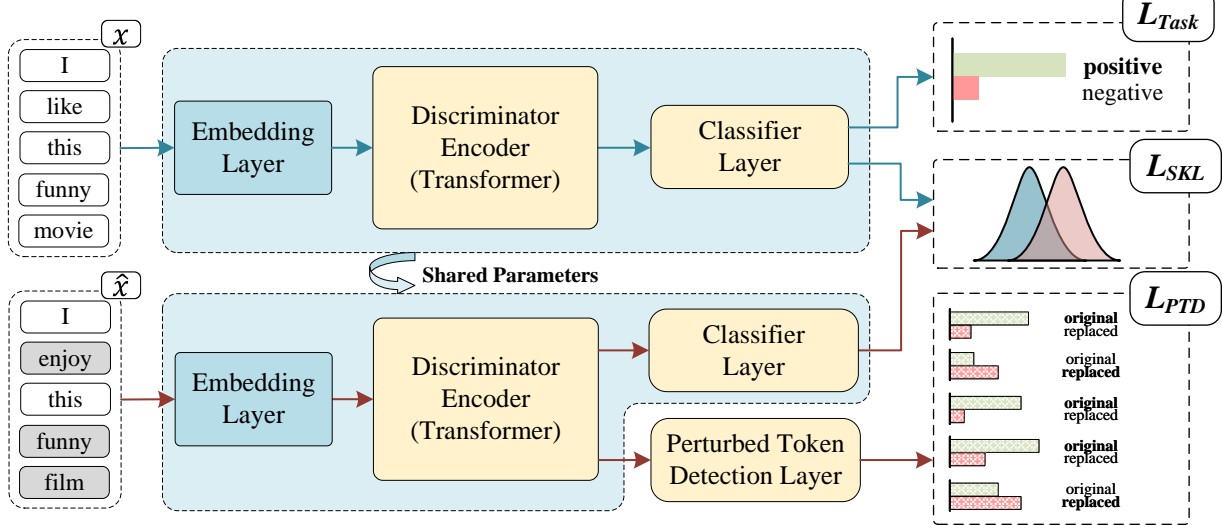

Figure 2: Overview of adversarial training process. By utilizing the generated adversarial examples in attack, the training process detects the perturbed tokens in the adversarial examples, and also regularizes the model behavior between clean and adversarial examples.

where $\mathcal{C}$ is set of masked token positions, $\hat{\boldsymbol{x}} = [\hat{x}_1, \hat{x}_2, ..., \hat{x}_n]$ is the generated adversarial example corresponding to $\boldsymbol{x}$.

## 3.3 Adversarial Training Process

Figure 2 illustrates the adversarial training process. After generating adversarial examples, the discriminator is trained using adversarial regularization and perturbed token detection techniques, which improves the robustness of the model.

Firstly, same as the standard fine-tuning, the discriminator is trained with cross-entropy loss to perform the classification task:

$$\mathcal{L}_{\text{Task}} = \text{CE}(p_{\theta_D}(\boldsymbol{x}), y) = -\sum_i^C y_i \log\left(p_{\theta_D}(\boldsymbol{x})\right)$$
(7)

where $\theta_D$ denotes the parameter of the discriminator.

**Perturbed Token Detection**  With the adversarial example $\hat{\boldsymbol{x}}$ is fed into the discriminator, a perturbed token detection layer is trained to classify each token in $\hat{\boldsymbol{x}}$ as either *original* or *replaced*.

$$\mathcal{L}_{\text{PTD}} = \mathbb{E}\left(-\sum_i \log p_{\theta_D}\left(\mathbb{1}\left(\hat{x}_i = x_i\right) \mid \hat{\boldsymbol{x}}, i\right)\right)$$
(8)

where $\mathbb{1}(\cdot)$ is the indicator function, $\hat{\boldsymbol{x}}$ is the adversarial example constructed with Eq.(6).

With $\mathcal{L}_{\text{PTD}}$, the model learns to distinguish intentionally misleading tokens. This token-level supervision facilitates robustness and enhances the efficiency of sample utilization.

**Adversarial Regularization**  To enhance adversarial robustness, we employ the symmetric KL-divergence as the adversarial regularization term, which promotes consistency in model performance between original $\boldsymbol{x}$ and adversarial examples $\hat{\boldsymbol{x}}$.

$$\begin{aligned}\mathcal{L}_{\text{SKL}} = {}& \mathcal{D}_{\text{KL}}(p_{\theta_D}(\boldsymbol{x})|p_{\theta_D}(\hat{\boldsymbol{x}})) \\ & + \mathcal{D}_{\text{KL}}(p_{\theta_D}(\hat{\boldsymbol{x}})|p_{\theta_D}(\boldsymbol{x}))\end{aligned}$$
(9)

where $\mathcal{D}_{\text{KL}}(\cdot|\cdot)$ is the KL-divergence between two distributions.

**Learning and Optimization**  The trainable parameters in the discriminator include the embedding layer, discriminator encoder, classifier layer, and perturbed token detection layer. The classifier layer is implemented as a shallow MLP, while the perturbed token detection layer is also implemented as a shallow MLP. The discriminator encoder consists of a stack of transformers.

Towards adversarial robustness, the overall loss is obtained by integrating Eq.(7), Eq.(8) and Eq.(9):

$$\mathcal{L} = \mathcal{L}_{\text{Task}} + \lambda_{\text{PTD}}\mathcal{L}_{\text{PTD}} + \lambda_{\text{SKL}}\mathcal{L}_{\text{SKL}} \quad (10)$$

where $\lambda_{\text{PTD}}$ and $\lambda_{\text{SKL}}$ are hyperparameters that control the balance between perturbed token detection and adversarial regularization. The hyperparameter search is provided in Appendix E.

The training objective of the discriminator is to minimize the overall loss $\mathcal{L}$. During the adversarial generative attack, the parameters are not updated. The attack is responsible for generating adversarial examples. And both the attack and training processes are jointly applied within each batch. To enhance training efficiency, parameters in GenerAT are initialized using pre-trained weights from the discriminative PLM.

## 4  Experiments

To evaluate the effectiveness of our framework, we compare GenerAT with the SoTA methods on five tasks. We also conduct additional experiments on model parameter analysis and provide a case study.

### 4.1  Tasks and Datasets

We conduct our experiments on AdvGLUE (Wang et al., 2021b), the most representative and widely used robustness evaluation benchmark. It consists of five challenging tasks in GLUE (Wang et al., 2018): Sentiment Analysis (SST-2), Duplicate Question Detection (QQP), and Natural Language Inference (NLI, including MNLI, RTE, and QNLI). In the construction of adversarial examples, AdvGLUE applies 14 textual adversarial attacks including various word-level, sentence-level perturbations, and human-crafted examples. The constructed adversarial examples are validated by human annotators. In experiments, we employ the development set of AdvGLUE since its test set is not publicly available.

### 4.2  Baselines and Evaluation Metric

We evaluate the performances of GenerAT by comparing it with the state-of-the-art (SoTA) adversarial training methods, robust fine-tuning methods and large language models (LLMs).

- **Adversarial training:** FreeLB (Zhu et al., 2020) adds adversarial perturbations to embedding and minimizing the risk in different regions. CreAT (Wu et al., 2023) is an adversarial training that finds perturbations in contextual representation.

- **Robust fine-tuning methods:** R3F (Aghajanyan et al., 2021) applies noise to the original pre-trained representations with regularization. ChildTuning (Xu et al., 2021) masks a subset of parameters and only updates the child network. Match-Tuning (Tong et al.,

2022) adds regularization between examples in the same batch.

- **LLMs:** BART-L (Lewis et al., 2020), GPT-J-6B (Wang, 2021), Flan-T5-L (Chung et al., 2022), GPT-NEOX-20B (Black et al., 2022), OPT-66B (Zhang et al., 2022), BLOOM (Scao et al., 2022), GPT-3 (text-davinci-002 and text-davinci-003) and ChatGPT.

Following the convention (Zhu et al., 2020; Li and Qiu, 2021; Li et al., 2021; Pan et al., 2022; Wu et al., 2023), the accuracy on adversarial examples is the metric for robustness, with higher accuracy indicates better robustness.

### 4.3  Implementation Details

We use DeBERTa-v3-large [1] as the pre-trained discriminative language model. We train our model on a single V100 GPU. Details of hyperparameters and training costs are provided in Appendix E. The results of comparative methods are based on the results reported in Tong et al. (2022); Wu et al. (2023); Wang et al. (2023a).

Our codes are provided in `https://github.com/Opdoop/GenerAT`.

### 4.4  Main Results

Table 1 gives the adversarial robustness results of our framework and the baselines on the AdvGLUE benchmark. We can see from the table that across all tasks, GenerAT significantly outperforms comparative baselines by a large margin.

In comparison to adversarial training and robust fine-tuning approaches, GenerAT exhibits a significant enhancement in adversarial robustness, nearly doubling the accuracy on adversarial examples. For instance, the SoTA of these methods (Match-Tuning) only achieves an average accuracy of 45.7, while GenerAT achieves an average accuracy of 80.1. This illustrates that adding perturbations to continuous embedding layer is sub-optimal in text domain, as the real attack adds perturbations at discrete tokens. GenerAT bridges this gap between continuous embedding representation and discrete tokens, which significantly improves the adversarial robustness of the model.

Compared to autoregressive language models, it is observed that increasing the parameter size slightly improves the model's robustness. For instance, transitioning from GPT-J-6B to GPT-3 (text-

---

[1]https://huggingface.co/microsoft/deberta-v3-large/

| Method | advSST-2 | advQQP | advMNLI-m | advQNLI | advRTE | Avg |
|---|---|---|---|---|---|---|
| *Adversarial Training Methods with BERT-base Model* (Wu et al., 2023) | | | | | | |
| Vanilla Fine-tuning (110 M) | 32.3 | 50.8 | 32.6 | 40.1 | 37.0 | 38.6 |
| FreeLB (110 M) | 31.6 | 51.0 | 33.5 | 45.4 | 42.0 | 40.7 |
| BERT MLM (110 M) | 32.0 | 48.5 | 27.6 | 43.4 | 45.9 | 39.5 |
| BERT CreAT (110 M) | 35.3 | 51.5 | 36.0 | 44.8 | 45.2 | 42.6 |
| *Robust Fine-tuning Methods with BERT-large Model* (Tong et al., 2022) | | | | | | |
| Vanilla Fine-tuning (340 M) | 47.6 | 38.5 | 35.0 | 46.4 | 41.7 | 41.8 |
| R3F (340 M) | 38.5 | 40.6 | 35.8 | 47.5 | 50.1 | 42.5 |
| ChildTuning$_F$ (340 M) | 34.5 | 40.4 | 33.9 | 47.5 | 42.0 | 39.6 |
| ChildTuning$_D$ (340 M) | 39.2 | 40.7 | 34.1 | 49.6 | 46.2 | 41.9 |
| Match-Tuning (340 M) | 51.4 | 41.5 | 35.5 | 47.5 | 52.5 | 45.7 |
| *State-of-the-art Large Language Models* (Wang et al., 2023a) | | | | | | |
| BART-L (407 M) | 43.9 | 37.2 | 41.3 | 48.0 | 43.2 | 42.7 |
| GPT-J-6B (6 B) | 51.3 | 41.0 | 26.4 | 50.0 | 43.2 | 42.4 |
| Flan-T5-L (11 B) | 59.5 | 41.0 | 51.2 | 50.0 | 43.2 | 49.0 |
| GPT-NEOX-20B (20 B) | 47.3 | 43.6 | 40.5 | 46.0 | 51.9 | 45.9 |
| OPT-66B (66 B) | 52.4 | 46.1 | 39.7 | 47.3 | 42.0 | 45.5 |
| BLOOM (176 B) | 51.3 | 41.0 | 26.4 | 50.0 | 43.2 | 42.4 |
| text-davinci-002 (175 B) | 54.0 | 71.8 | 45.4 | 54.7 | 64.2 | 58.0 |
| text-davinci-003 (175 B) | 55.4 | 44.9 | 55.4 | 61.5 | 65.4 | 56.5 |
| ChatGPT (175 B) | 60.1 | 82.0 | 67.8 | 65.5 | 75.3 | 70.1 |
| *Robust Training Methods with DeBERTa-v3-large Model* | | | | | | |
| GenerAT (436 M) | **69.6** | **89.7** | **78.5** | **73.6** | **88.9** | **80.1** |

Table 1: Adversarial robustness results on the AdvGLUE benchmark. We report the accuracy values on adversarial examples. The best performing scores are in **bold**. Models are ranked by parameter size. Avg stands for the average accuracy on AdvGLUE. Our GenerAT surpasses the baselines by a large margin.

davinci-002) leads to an average accuracy increase from 42.4 to 58.0. However, models with the same parameter size exhibit considerable performance variation, with BLOOM only achieving 42.4 in average accuracy. Notably, the Encoder-decoder structure of Flan-T5-L, incorporating instruction tuning, outperforms other autoregressive language models of similar scale, such as GPT-NEOX-20B, underscoring the importance of model architecture for adversarial robustness. Among LLMs, Chat-GPT achieves the highest average accuracy of 70.1. GenerAT utilizes DeBERTa-v3-large as its base PLM. With less than 1% of parameters compared to ChatGPT, GenerAT surpasses ChatGPT by 10% in average accuracy. This result demonstrates the effectiveness of our approach. Table 2 gives the results of GenerAT on clean GLUE dev datasets. It can be seen from the table that, in contrast to classical adversarial training, our GenerAT framework does not sacrifice the performance on clean data.

## 4.5 Results on Ablation Study

To evaluate the effectiveness of GenerAT, we conduct the ablation study on its different variances:

**- PTD:** removing the perturbed token detection (PTD) module from GenerAT.

**- GAA:** removing the generative adversarial attack (GAA) and $\mathcal{L}_{SKL}$ from GenerAT.

**- GAA - PTD:** removing both GAA and PTD modules from GenerAT.

Table 3 gives ablation results on different components of GenerAT in advGLUE datasets. It can be seen from the table that the individual modules of GAA and PTD each play a pivotal role in enhancing adversarial robustness. Specifically, when GAA is removed, the average accuracy declines to 73.7. Similarly, in the absence of PTD, the average accuracy declines to 71.1. When both GAA and PTD are removed simultaneously, the model

| GLUE dev | SST-2 | QQP | MNLI-m | MNLI-mm | QNLI | RTE | Avg |
|---|---|---|---|---|---|---|---|
| GenerAT | 96.3 | 89.7 | 91.4 | 91.5 | 95.5 | 89.9 | 92.4 |

Table 2: Accuracy results of GenerAT on clean GLUE dev datasets. MNLI-m is the matched version of MNLI and MNLI-mm is the mismatched version of MNLI.

| Finetuning | GAA | PTD | advSST-2 | advQQP | advMNLI-m | advQNLI | advRTE | Avg |
|---|---|---|---|---|---|---|---|---|
| ✓ | ✗ | ✗ | 59.2 | 69.3 | 64.2 | 63.2 | 79.0 | 66.9 |
| ✓ | ✓ | ✗ | 62.1 | 75.6 | 68.5 | 65.5 | 83.9 | 71.1 |
| ✓ | ✗ | ✓ | 65.4 | 78.2 | 71.9 | 67.8 | 85.2 | 73.7 |
| ✓ | ✓ | ✓ | 69.6 | 89.7 | 78.5 | 73.6 | 88.9 | 80.1 |

Table 3: Ablation results on different components of GenerAT in advRTE dataset.

| Base Model | # Layers | Parameters | advSST-2 | advQQP | advMNLI-m | advQNLI | advRTE | Avg |
|---|---|---|---|---|---|---|---|---|
| DeBERTa-v3-large | 24 | 436M | 69.6 | 89.7 | 78.5 | 73.6 | 88.9 | 80.1 |
| DeBERTa-v3-xsmall | 12 | 71M | 52.0 | 50.4 | 61.7 | 56.8 | 61.5 | 56.5 |

Table 4: Effect of parameter scales on GenerAT, where Avg stands for the average accuracy on AdvGLUE datasets.

degrades to vanilla finetuning and the average accuracy declines to 66.9. This indicates that both GAA and PTD are important for enhancing model robustness. The results of the ablation study further verify the effectiveness of each component in our framework.

### 4.6 The Effect of Parameter Size

In order to assess the impact of parameter size on the performance of GenerAT, we conducted an evaluation utilizing a reduced version of DeBERTa. The average performance on AdvGLUE of GenerAT based on *DeBERTa-v3-xsmall* and *DeBERTa-v3-large* are shown in Table 4. We observe that downscale the parameter size from 436M to 71M, the average accuracy drops from 80.1 to 56.5, which suggests a larger base model is important for robustness. The experimental findings reveal that employing larger parameter sizes in conjunction with deeper network architectures markedly improves the model's performance in the presence of adversarial examples and highlight the importance of having a sufficiently large parameter space to learn robust representations. And DeBERTa-v3-large is a suitable choice in this regard.

We also visualize the relationship between model size and performance in Figure 3. We can see that GenerAT based on *DeBERTa-v3-xsmall* achieves comparable robustness to GPT-3. GenerAT based on *DeBERTa-large* achieves nearly twice the robustness compared to models with similar parameter scales, such as BERT-large. Moreover, GenerAT also surpasses other LLMs like

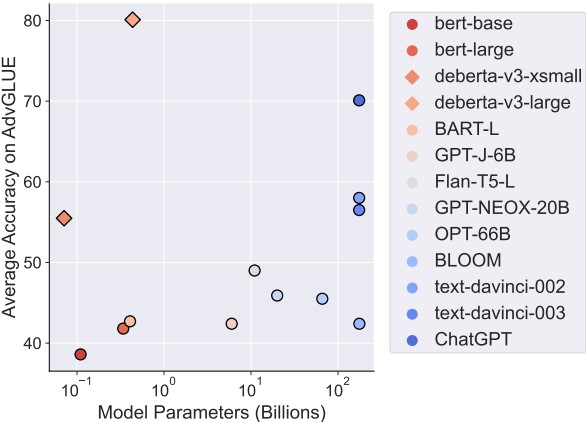

Figure 3: Illustration of model robustness results.

ChatGPT. These findings suggest that simply increasing the model size does not inherently mitigate the issue of adversarial vulnerability. Instead, the structure and training strategy of the model employed in downstream tasks play a crucial role in adversarial robustness.

### 4.7 Case Study: Perturbed Token Detection

We further visualize the perturbed token detection results on adversarial examples in Table 5. In the table, perturbed positions are marked in the original example with gray shadows and the detection results of the corresponding adversarial example are marked with yellow shadows. The shade of the color represents the normalized perturbed probability. For instance, in the advSST-2 example, the word 'wildly' in the original example is replaced with 'strangely' in the adversarial example. The

| Dataset | Type | Text Content |
|---------|------|--------------|
| advSST-2 | Original Example | ' s too much forced drama in this wildly uneven movie , about a young man ' s battle with his inescapable past and uncertain future in a very sha pable but largely un fulfilling present |
| | Adversarial Example | ' s so much forced drama in this strangely uneven movie , about a young man ' s battles with his inescapable past and uncertain future in the very sha pable but often un fulfilling present |
| advQQP | Original Example | What would happen if an astronaut dies in space while on ISS ? [SEP] What would it be like if an astronaut died aboard the International Space Station ? |
| | Adversarial Example | What would happen if one astronaut dies in space while on Mars ? [SEP] What would it feel like if an astronaut died within the International Flight Station ? |
| advMNLI-m | Original Example | [CLS] Because of the limited money available , the first grants were restricted to funding for civil legal services and hotline s . [SEP] Due to limited money available companies had to cut spending |
| | Adversarial Example | [CLS] Because of scarce limited money available , the first efforts were restricted in funding for civil legal services and hotline support . [SEP] Due to inadequate money available companies had to cut spending |
| advQNLI | Original Example | A large - scale solar distillation project was first constructed in 1872 in the Chilean mining town of Las Salinas . [SEP] When was the first large solar distillation plant created ? |
| | Adversarial Example | A large - scale water distillation project was first established in 1872 near the Chilean mining town of Las Salinas . [SEP] When was the first large solar distillation plant constructed ? |
| advRTE | Original Example | Companies are working to reduce the interval between drug discovery and marketing . [SEP] Companies are working to shorten the new drug development period to an average of eight to nine years in the US . |
| | Adversarial Example | Companies are working to shorten the roadblocks between drug discovery and marketing . [SEP] Investigators are working to shorten the new drug development period from an average of eight to nine years in the US . |

Table 5: Qualitative results of perturbed token detection. In the original example, the token position of perturbations are marked in gray. In the adversarial example, the detection results are marked in yellow. The darker the color, the more likely the token is replaced.

darker the color, the more likely the token is replaced. We can see from the table that perturbed token detection is skeptical of most tokens and discovers the real perturbed tokens with high confidence. This indicates that further enhancing the accuracy of perturbed token detection may lead to the increase of robustness. Thus the ability to detect perturbed tokens can provide valuable token-level supervision signals, making it a crucial task for improving model robustness.

## 5 Conclusion

This paper presents a novel generative adversarial training framework to bridge the gap between embedding representations and discrete text tokens in existing adversarial training methods. Our proposed framework integrates gradient-based learning, adversarial example generation, and perturbed token detection for improving adversarial robustness. The generative adversarial attack shares the embeddings between the classifier and the generative model, which allows the generative model to utilize gradients from the classifier for generating perturbed tokens. Then the adversarial training process incorporates perturbed token detection into adversarial regularization, which provides token-level supervision and enhances sample usage efficiency. Extensive experiments on five datasets in AdvGLUE benchmark demonstrate that our framework significantly boosts the model robustness.

## Limitations

In our generative adversarial attack, we omit semantic constraints for the sake of efficiency. Although some prior research suggests that semantic constraints are important for preserving the original semantic meaning when generating adversarial perturbations, we observed that the automatic constraint metric is sensitive across different datasets. And establishing an appropriate threshold for semantic constraints often necessitates significant human involvement to ensure rationality. However, our experimental results show that even without explicit semantic constraints, our approach is still effective. The impact of semantic constraints is left for future research.

As we focus on adversarial robustness in downstream tasks, the evaluation of our GenerAT framework has been limited to robustness benchmarks. The potential of this framework in scenarios involving data shift, out-of-distribution samples, and other situations remains a topic for future research.

## Ethics Statement

Our work focuses on enhancing the adversarial robustness of models in downstream tasks. The proposed GenerAT framework has demonstrated significant improvements in adversarial robustness. Careful investigations are needed to understand the impact of enhancing robustness on existing biases and fairness issues in machine learning systems.

## Acknowledgments

This work is supported in part by the Ministry of Science and Technology of China under Grant #2020AAA0108401, and the National Natural Science Foundation of China under Grants #62206287, #11832001 and #72293575.

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

| Model | Architecture | | | | Pre-training Task | | | | | |
|---|---|---|---|---|---|---|---|---|---|---|
| | Encoder | Encoder-Decoder | Discriminator | Decoder | MLM | SD | RTD | NSP | CLM | IF |
| BERT | ✓ | | | | ✓ | | | ✓ | | |
| T5 | | ✓ | | | | ✓ | | | | |
| FLAN-T5 | | ✓ | | | | ✓ | | | | ✓ |
| ELECTRA | ✓ | | ✓ | | ✓ | | ✓ | | | |
| DeBERTa V1 | ✓ | | | | ✓ | | | | | |
| DeBERTa V3 | ✓ | | ✓ | | ✓ | | ✓ | | | |
| GPT | | | | ✓ | | | | | ✓ | |
| OPT | | | | ✓ | | | | | ✓ | |
| BLOOM | | | | ✓ | | | | | ✓ | |

Table 6: Comparison of different language models.

| Model | advSST-2 | advQQP | advMNLI-m | advQNLI | advRTE | Avg |
|---|---|---|---|---|---|---|
| gpt-3.5-turbo (zero-shot) | 66.9 | 29.5 | 63.6 | 43.9 | 39.5 | 48.7 |
| gpt-3.5-turbo (five-shot) | 75.5 | 35.6 | 56.9 | 43.4 | 35.5 | 49.4 |
| gpt-3.5-turbo (ten-shot) | 76.8 | 39.7 | 55.9 | 44.9 | 40.9 | 51.6 |
| GenerAT | 69.6 | 89.7 | 78.5 | 73.6 | 88.9 | 80.1 |

Table 7: Robustness results on AdvGLUE of GPT-3.5

## A  Comparison between LLMs

Table 6 provides a comprehensive comparison of different language models, with a primary focus on the analysis of architecture and pre-training tasks, both of which have a significant impact on the acquired representations. In terms of architecture, different designs are considered, including Encoder-only, Decoder-only, Encoder-decoder, and Encoder with an additional Discriminator. The choice of pre-training task is closely aligned with the selected architecture. Specifically, the pre-training task options encompass Masked Language Modeling (MLM), Span-Mask Denoising (SD), Replaced Token Prediction (RTD), Next Sentence Prediction (NSP), Causal Language Modeling (CLM) and Instruction Finetuning (IF). Our proposed generative adversarial training framework GenerAT builds on the DeBERTa-v3, where the discriminator is trained using the RTD and the generator is trained using MLM. In contrast to MLM or CLM approaches that rely on contextual information for representation learning, we argue that the discriminator serves as a superior backbone network for adversarial robustness, as it acquires representations by discerning similar words. Therefore, we select DeBERTa-v3 as the backbone network of our framework.

## B  Robustness of GPT-3.5

Table 7 shows the results of GPT-3.5. We use the prompt provided in (Wang et al., 2023b) with randomly selected examples. It can be seen from the table that few-shot examples marginally improve the robustness of GPT-3.5.

## C  Robustness under Textfooler Attack

| Method | IMDB | AG NEWS |
|---|---|---|
| BERT | 2.8 | 19.4 |
| RoBERTa | 25.2 | 25.2 |
| Adv-HotFlip (BERT) | 8.0 | 18.2 |
| FreeLB (BERT) | 7.3 | 20.1 |
| FreeLB++ (BERT) | 45.3 | - |
| RanMASK (RoBERTa) | 23.7 | - |
| TAVAT | 27.6 | 39.7 |
| InfoBERT | 27.4 | 29.2 |
| Flooding | 39.5 | 38.8 |
| Flooding-X | 40.5 | 42.4 |
| Text Purification(BERT) | 51.0 | 34.9 |
| Text Purification(RoBERTa) | 54.3 | 34.2 |
| GenerAT | 75.6 | 44.0 |

Table 8: Robustness Results under Textfooler Attack on IMDB and AG NEWS datasets.

Table 8 gives the results under Textfooler attack. We focus on IMDB and AGNEWS for experimentation (as SST-2 and QNLI have been tested under multiple attacks in AdvGLUE). The results of baselines are derived from (Li et al., 2023) and (Liu et al., 2022b). It can be seen that our generative adversarial training method GenerAT still surpasses the compared baselines by a large margin.

## D  Additional Results

To ensure fairness and consistency in comparison, Table 1 only presents those results that all the baselines have reported. Here, we provide additional experiment results in Table 9.

| Method | advQQP (F1) | advMNLI-mm (accuracy) |
|---|---|---|
| *Adversarial Training Methods with BERT-base Model* | | |
| Vanilla Fine-tuning (110 M) | - | 19.3 |
| FreeLB (110 M) | - | 21.9 |
| BERT MLM (110 M) | - | 20.8 |
| BERT CreAT (110 M) | - | 22.0 |
| *Robust Fine-tuning Methods with BERT-large Model* | | |
| Vanilla Fine-tuning (340 M) | 27.59 | 30.00 |
| R3F (340 M) | 35.23 | 30.26 |
| ChildTuning$_F$ (340 M) | 35.82 | 26.53 |
| ChildTuning$_D$ (340 M) | 39.80 | 27.84 |
| Match-Tuning (340 M) | 32.62 | 31.07 |
| *State-of-the-art Large Language Models* | | |
| BART-L (407 M) | - | - |
| GPT-J-6B (6 B) | - | - |
| Flan-T5-L (11 B) | - | - |
| GPT-NEOX-20B (20 B) | - | - |
| OPT-66B (66 B) | - | - |
| BLOOM (176 B) | - | - |
| text-davinci-002 (175 B) | - | - |
| text-davinci-003 (175 B) | - | - |
| ChatGPT (175 B) | - | - |
| *Robust Training Methods with DeBERTa-v3-large Model* | | |
| GenerAT | 86.95 | 70.52 |

Table 9: Results on advQQP and advMNLI-mm.

# E  Model Hyperparameters

Table 10 provides the hyper-parameters for the best performance of GenerAT on each advGLUE dataset. In hyper-parameter search process, the select space of $\lambda_{PTD}$ is {0.5,1,1.5,2,2.5} and the select space of $\lambda_{SKL}$ is {1.5,3.5,4,4.5,5,6,6.5,7}. $\lambda_{PTD}$ and $\lambda_{SKL}$ are selected on the advRTD dataset, and the best hyperparameter values are subsequently applied to the other datasets. The learning rate is set within the range of {1e-5,2e-6,5e-6,7e-6}. The total training time is calculated based on a single V100 (32GB) GPU. For the complete parameter setup, please refer to our code.

| Hyper-parameter | advSST-2 | advQQP | advMNLI-m | advQNLI | advRTE |
|---|---|---|---|---|---|
| Number of Layers | 24 | 24 | 24 | 24 | 24 |
| Hidden Size | 1024 | 1024 | 1024 | 1024 | 1024 |
| Attention Heads | 16 | 16 | 16 | 16 | 16 |
| Attention Head Size | 64 | 64 | 64 | 64 | 64 |
| Dropout | 0.1 | 0.1 | 0.1 | 0.1 | 0.1 |
| Learning Rate | 2e-6 | 1e-5 | 7e-6 | 7e-6 | 7e-6 |
| Batch Size | 16 | 8 | 8 | 16 | 8 |
| Epoch | 2 | 2 | 2 | 3 | 3 |
| $\lambda_{PTD}$ | 1.5 | 1.5 | 1.5 | 1.5 | 1.5 |
| $\lambda_{SKL}$ | 4 | 4 | 4 | 4 | 4 |
| Training Time (Total, hh:mm) | 01:57 | 43:43 | 25:52 | 09:54 | 00:17 |

Table 10: Hyper-parameters of GenerAT on AdvGLUE.