# OpenReview forum: "Generative Adversarial Training with Perturbed Token Detection for Model Robustness"
_EMNLP/2023/Conference — EMNLP 2023 Main_

### Official Review · Reviewer_1YBd · 2023-08-02

**Soundness:** 4

**Excitement:**

4: Strong: This paper deepens the understanding of some phenomenon or lowers the barriers to an existing research direction.

**Missing References:**

Ni, Shiwen, Jiawen Li, and Hung-Yu Kao. "Dropattack: A masked weight adversarial training method to improve generalization of neural networks." arXiv preprint arXiv:2108.12805 (2021).

**Paper Topic And Main Contributions:**

The topic of this thesis is Generative Adversarial Training with Perturbation Labeling Detection for Model Robustness. The main contribution of the paper is to propose a novel generative adversarial training framework to improve model robustness by integrating gradient learning, adversarial sample generation and perturbation labeling detection. The framework significantly improves the robustness of the model on multiple datasets and exceeds the recent results of ChatGPT. This research fills the gap between continuous perturbation and discrete text markers in existing adversarial training methods and generates discrete perturbation markers via gradient propagation, combined with perturbation marker detection and adversarial regularization to improve sample utilization efficiency.

**Reasons To Accept:**

1. This paper proposes a novel generative adversarial training framework to improve model robustness by integrating gradient learning, adversarial sample generation, and perturbation marker detection.

2. This paper addresses the gap between continuous perturbation and discrete text labeling in existing adversarial training methods, which has long been a limiting factor.

3. Extensive experiments on five datasets demonstrate the effectiveness of the proposed generative algorithm training framework, which significantly improves robustness.


**Reasons To Reject:**

The proposed method is almost the same as the training method for the model ELECTRA.

The authors may also need to test whether the performance of the proposed method is degraded on non-adversarial datasets.

**Reproducibility:**

4: Could mostly reproduce the results, but there may be some variation because of sample variance or minor variations in their interpretation of the protocol or method.

**Reviewer Confidence:**

4: Quite sure. I tried to check the important points carefully. It's unlikely, though conceivable, that I missed something that should affect my ratings.

---

### Official Review · Reviewer_LmDX · 2023-08-03

**Soundness:** 4

**Excitement:**

4: Strong: This paper deepens the understanding of some phenomenon or lowers the barriers to an existing research direction.

**Missing References:**

please see the reasons above

**Paper Topic And Main Contributions:**

This paper proposes a generative framework to detect adversarial text against adversarial attacks. They propose a framework that utilizes gradient-based adv. training and uses the gradients to guide the generation process sothat the model robustness is improved.

**Questions For The Authors:**

please see the reasons above.

**Reasons To Accept:**

1. the motivation is strong. it is very intuitive to consider using generative models to detect/purify adversaries in improving model robustness. this paper proposes a framework to accomplish this goal
2. the method is reasonable: the adversarial training can guide the generative process so that the model can detect the adversaries and therefore improve model robustness against adversaries
3. the experiments show the effectiveness of the method

**Reasons To Reject:**

1. some baseline methods should be compared with this work. e.g.  [1] Searching for an Effective Defender: Benchmarking Defense against Adversarial Word Substitution [2] Certified robustness to text adversarial attacks by randomized [mask] [3] Text Adversarial Purification as Defense against Adversarial Attacks. the purification process used in the image domain is somehow similar to the proposed idea, therefore, it is important to consider these lines of works.
2. the tests use static datasets while it is somewhat important to see how the proposed framework can actually help resist adversarial attack process such as textfooler.
3. how do you instruct the ChatGPT models is important, I'm concerned that with some carefully designed instructs, it is possible that ChatGPT can obtain significant improvements.

**Reproducibility:**

4: Could mostly reproduce the results, but there may be some variation because of sample variance or minor variations in their interpretation of the protocol or method.

**Reviewer Confidence:**

4: Quite sure. I tried to check the important points carefully. It's unlikely, though conceivable, that I missed something that should affect my ratings.

---

### Official Review · Reviewer_WUQd · 2023-08-04

**Soundness:** 4

**Excitement:**

3: Ambivalent: It has merits (e.g., it reports state-of-the-art results, the idea is nice), but there are key weaknesses (e.g., it describes incremental work), and it can significantly benefit from another round of revision. However, I won't object to accepting it if my co-reviewers champion it.

**Missing References:**

[1] Linyang Li and Xipeng Qiu. 2021. Token-aware virtual adversarial training in natural language understanding.
[2] Boxin Wang, Shuohang Wang, Yu Cheng, Zhe Gan, Ruoxi Jia, Bo Li, and Jingjing Liu. 2021a Infobert: Improving robustness of language models from an information theoretic perspective.
[3] Qin Liu, Rui Zheng, Bao Rong, Jingyi Liu, ZhiHua Liu, Zhanzhan Cheng, Liang Qiao, Tao Gui, Qi Zhang, and Xuanjing Huang. 2022. Flooding-X: Improving BERT’s resistance to adversarial attacks via loss restricted fine-tuning.


**Paper Topic And Main Contributions:**

This paper first proposes a novel generative adversarial training framework, which bridges the robustness gap of continuous embedding representations discrete text tokens, boosting the robustness of model. In generative adversarial attack, the embeddings are shared to leverage the gradients from the classifier for generating perturbed tokens. Then, adversarial training combines adversarial regularization with perturbed token detection to provide token-level supervision.

**Questions For The Authors:**

1. In line 305 and 317, the authors should explain the mean of eta.
2. Adversarial training has other method training in embedding representations, such as TAVAT[1], InfoBERT[2], Flooding-X[3]. The authors should compare their method with these methods as well.
3. The authors should evaluate their methods in BERT-base and BERT-large model instead of DeBERTa model, which is more fairer and more convincing.
[1] Token-aware virtual adversarial training in natural language understanding.
[2] Infobert: Improving robustness of language models from an information theoretic perspective.
[3] Flooding-X: Improving BERT’s resistance to adversarial attacks via loss restricted fine-tuning.


**Reasons To Accept:**

1. This paper proposes a novel generative adversarial training framework to boost the robust performance of model.
2. The proposed method outperform the baseline with a clear margin and it can detect perturbed token accurately.


**Reasons To Reject:**

1. The authors do not analysis the weakness of training with continuous representations of perturbations and the strength of the proposed method.
2. The comparison of LLMs and proposed method is not fair. The LLMs are tested in zero-shot setting while this proposed method are trained by adversarial examples before evaluating.
3. DeBERTa model is different from BERT-base model and the former has better performance than the latter. The author should compare the results with same BERT model.


**Reproducibility:**

4: Could mostly reproduce the results, but there may be some variation because of sample variance or minor variations in their interpretation of the protocol or method.

**Reviewer Confidence:**

4: Quite sure. I tried to check the important points carefully. It's unlikely, though conceivable, that I missed something that should affect my ratings.

---

### Meta-Review · Area_Chair_QhW2 · 2023-09-18

**Recommendation:** 5

**Metareview:**

This paper presents a generative adversarial training framework that combines three components: gradient-based learning, adversarial example generation, and perturbed token detection. The experiments demonstrate the effectiveness of the proposed method on five benchmark datasets. The reviewers acknowledged the contribution and merit of this work, including the motivation of the research, the proposed method, and empirical evidence from experiments. However, there are also issues with the experiment design, especially the baselines and additional result analysis.

---

### Decision · Program_Chairs · 2023-10-07

**Decision:**

Accept-Main

**Comment:**

This paper presents a generative adversarial training framework that combines three components: gradient-based learning, adversarial example generation, and perturbed token detection. The experiments demonstrate the effectiveness of the proposed method on five benchmark datasets. The reviewers acknowledged the contribution and merit of this work, including the motivation of the research, the proposed method, and empirical evidence from experiments. However, there are also issues with the experiment design, especially the baselines and additional result analysis.